# Gut Microbiome-Host Genetics Co-Evolution Shapes Adiposity by Modulating Energy and Lipid Metabolism in Selectively Bred Broiler Chickens

**DOI:** 10.3390/ani14223174

**Published:** 2024-11-06

**Authors:** Guangqi Gao, Yangbo Jiao, Lai-Yu Kwok, Zhi Zhong

**Affiliations:** 1Key Laboratory of Dairy Biotechnology and Engineering, Ministry of Education, Inner Mongolia Agricultural University, Hohhot 010018, China; guangqigao@163.com (G.G.); jiaoyangbo@emails.imau.edu.cn (Y.J.); kwok_ly@yahoo.com (L.-Y.K.); 2Key Laboratory of Dairy Products Processing, Ministry of Agriculture and Rural Affairs, Inner Mongolia Agricultural University, Hohhot 010018, China; 3Inner Mongolia Key Laboratory of Dairy Biotechnology and Engineering, Inner Mongolia Agricultural University, Hohhot 010018, China

**Keywords:** *Gallus gallus*, selective breeding, fat deposition, single nucleotide polymorphism, gut-genome axis, apolipoprotein B, peroxisome proliferator-activated receptor-gamma, perilipin, *Lactobacillus*

## Abstract

Chicken is an important part of the human diet, and it is considered a nutritious and healthy food. Excessive abdominal fat deposition in broiler chickens presents a significant hurdle for both poultry producers and the food industry. This study compared the gut microbiome composition and host whole-genome profiles between fat-line and lean-line broiler chickens that had been selectively bred for divergent abdominal fat levels over 15 generations. Our findings suggest that the two chicken lines exhibited significant differences in their gut microbiota and the selective breeding process also led to genomic variations in the lean broilers, with single nucleotide polymorphisms predominantly observed in genes related to energy and lipid metabolism. This study offers new insights into the intricate gut-genome interactions in chicken fat metabolism, paving the way for more effective breeding and microbiome-based strategies to manage adiposity in poultry.

## 1. Introduction

Excessive abdominal fat deposition in broiler chickens presents a significant hurdle for both poultry producers and the food industry [1]. This surplus adiposity not only deteriorates the quality and desirability of the meat but also leads to elevated concentrations of residual drugs and harmful substances, posing a potential threat to food safety [2]. Furthermore, obesity in chickens leads to elevated energy demands and feed costs, thereby reducing the dressing percentage, a key metric of growth rate and meat yield [3]. These factors negatively affect the feed conversion ratio and overall farming profitability, ultimately undermining the economic sustainability of poultry farming [4]. Controlling abdominal fat deposition is, therefore, crucial for optimizing farming efficiency and enhancing the appeal of poultry products to consumers. Furthermore, reducing abdominal fat could enhance the quality and acceptance of poultry products in the marketplace.

The effective control of abdominal fat accumulation in broiler chickens requires a holistic approach encompassing factors such as rearing management, feed formulation, and physical activity promotion [5,6]. One promising strategy is the use of selective breeding to generate low-fat broiler varieties. Selection breeding can not only directly improve fat deposition in broilers but also enhance their overall health and growth efficiency through genetic improvement. For instance, Tan et al. comprehensively identifies genetic markers and regulatory mechanisms (such as the SOX6-MYH1s axis) involved in muscle development and yield in broiler chickens, offering new targets for selective breeding to enhance meat production efficiency and reduce myopathy [7]. Additionally, Shi et al. revealed the effects of selective breeding on skeletal muscle metabolism and meat quality through metabolomics technology, providing significant metabolic pathways and biomarkers for exploring differences in meat quality between different broiler populations [8]. This breeding approach involves identifying and selecting breeders with naturally lower levels of abdominal fat, and then systematically breeding their offspring to inherit the desired lean phenotype [9,10,11]. By selecting for reduced fat deposition over multiple generations, producers can establish stable, low-fat broiler lines that maintain the favorable trait. And by combining selective breeding for leanness with well-managed rearing practices, producers can effectively produce broiler chickens with reduced abdominal fat.

Alongside selective breeding, it is crucial to consider the impact of dietary and other environmental factors on the diversity and composition of the gut microbiota—a complex microbial community that is intricately tied to animal health and immunity [12,13]. Optimizing the nutritional composition and energy density of broiler feed can also help stimulate greater energy expenditure and mitigate excess fat accumulation [14]. Moreover, the varying carbohydrates, proteins, and fat sources of broiler’s diet have direct effects on the gut microbiome diversity, composition, and structure [15,16,17]. There is a close, bidirectional relationship between the gut microbiota and host fat deposition [18]. Emerging research suggests that the gut microbiota can actively modulate the host’s energy metabolism and fat deposition patterns, thereby influencing overall body weight and adiposity [19,20]. In fact, the gut microbiota appears to play an important role in regulating fat deposition, independent of host’s genetic makeup. For example, transplanting gut microbiota from lean to obese mice can significantly reduce body weight and fat tissue in the recipients [21]. Even among broiler chickens with the same genotype, differences in gut microbiota composition can result in varying patterns of fat deposition [22].

In our previous study, we utilized both a lean chicken line and a fat chicken line with divergent levels of adiposity, which were generated through 15 generations of selective breeding based on the plasma concentration of very low-density lipoprotein and abdominal fat percentage [23]. This selective breeding approach likely induced genomic variations that favored genotypes associated with reduced obesity and fat deposition, without directly manipulating the gut microbiota. Considering the important regulatory roles of gut microbiota and host genetics in fat metabolism, it becomes crucial to comprehend their reciprocal interactions in determining the extent of fat deposition. Our hypothesis posits that the changes in chicken gut microbiota after selective breeding could be associated with host genomic variations. By analyzing the correlation between genetic variations in broiler chickens and alterations in gut microbiota composition, this study aimed to shed light on the complex, co-evolutionary relationship between the host genome and the gut microbial ecosystem and how this interplay influenced host physiology, particularly in the context of fat deposition. Understanding these host–microbiome interactions could provide valuable insights for developing more holistic strategies to manage excessive abdominal fat accumulation in commercial broiler production.

## 2. Materials and Methods

### 2.1. Animals and Samples

This study was approved by the Special Committee on Scientific Research and Academic Ethics of Inner Mongolia Agricultural University (approval no. NND2023107). The white feather broiler chickens utilized in this study were generously supplied by Northeast Agricultural University. The broiler selection process involved 15 generations of breeding to distinguish between the fat-line, which was characterized by increased abdominal fat content, and the lean-line, which was identified by reduced abdominal fat content. The selective breeding process was primarily guided by monitoring the plasma concentration of very low-density lipoprotein and the percentage of abdominal fat. A total of 29 broilers were studied, comprising 14 from the fat-line (7 male and 7 female) and 15 from the lean-line (8 male and 7 female). The distribution of males and females was kept even. All chickens were derived from a single colony. Throughout various life stages, the chickens were provided with different diets as per the Arbor Acres Plus parent stock breeder management guide and nutrition specifications (http://en.aviagen.com/). During the sampling period (ages ranging from 37 to 40 weeks), the chickens were subjected to feed restriction on a standard diet comprising 14.2% crude protein and 2745 kcal/kg of metabolic energy. Table 1 details the nutritive content of the feed and the digestible amino acid supplementation at the time of sampling. Through the above rigorous quality control measures, we ensured the consistency of feed components and quality, thereby effectively eliminating potential interference from feed on the results of intestinal microbiome diversity studies.

Each chicken was individually housed in a dedicated cage to minimize potential contamination from uncontrolled particle intake and feathers. The temperature was maintained at a consistent level between 16 and 18 °C, and the humidity was kept at a level between 50 and 60%. The broilers were fed specific diets at different life stages following the guidelines set out in the Arbor Acres Plus Parent Stock Breeder Management Guide and Nutrition Specifications (http://en.aviagen.com/), with access to an ample water supply. The health of the birds and environmental conditions were inspected twice daily by trained personnel. Excreta and blood samples were collected at 40 days of age for microbiome metagenomic sequencing and host whole-genome sequencing, respectively. Briefly, 5–6 mL of blood was collected from a vein under the wing of the broiler, centrifuged at 3000 r/min for 10 min, and the serum was separated. The collected excreta and serum samples were stored at −80 °C until further analysis.

### 2.2. Metagenomic Shotgun Sequencing

The process of DNA sequencing analysis of excreta samples entailed a series of discrete steps. Firstly, collected excreta samples were thawed, and total genomic DNA was extracted using the QIAamp Fast DNA Stool Mini Kit (Qiagen, Hilden, Germany). Next, DNA libraries were constructed using the NEBNext^®^ Ultra™ DNA Library Prep Kit for Illumina (New England Biolabs, Ipswich, MA, USA), following the manufacturer’s instructions. This resulted in the generation of DNA fragments of approximately 300 base pairs. Paired-end reads of 150 base pairs in both the forward and reverse directions were generated by sequencing the samples on the Illumina HiSeq2000 platform (Illumina Inc., San Diego, CA, USA). To ensure the quality of the reads, low-quality reads were removed using the KneadData pipeline (version 0.7.5), a well-established quality control method available at http://huttenhower.sph.harvard.edu/kneaddata (accessed on 1 October 2024).

The clean metagenomics dataset was assembled into contigs using MEGAHIT software (version 1.2.9) (https://github.com/voutcn/megahit, accessed on 1 October 2024) with the default settings. Subsequently, the metagenomic species were annotated using MetaPhlAn2 (version 3.0.2) [24]. Functional annotation was achieved using the HUMAnN2 pipeline (version 0.11.2) [25], using the UniRef90 database (https://www.uniprot.org/help/uniref, accessed on 1 October 2024). These methods facilitated the taxonomic and functional characterization of the metagenomics dataset.

### 2.3. Whole-Genome Sequencing

Genomic DNA was extracted from the serum samples of broiler chickens using a commercial DNA extraction kit (DP210831, Tiangen, Beijing, China), following the manufacturer’s instructions. Subsequently, the DNA samples were fragmented by sonication, producing fragments of around 300 base pairs. These prepared libraries underwent next-generation sequencing on an Illumina HiSeq 4000 platform (Illumina Inc., San Diego, CA, USA). The resulting reads were filtered and aligned to the reference broiler genome sequence (ENSEMBL chicken release 67, available at http://may2012.archive.ensembl.org/Gallus_gallus/Info/Index, accessed on 1 October 2024) using the Burrows-Wheeler aligner (version 0.7.12) [26]. Duplicate reads were identified and excluded using the Picard tools (version 2.25.5) (https://broadinstitute.github.io/picard/, accessed on 1 October 2024). To minimize the occurrence of errors, reads aligned in proximity to indel regions were realigned, and the base quality scores were recalibrated. Single nucleotide polymorphisms (SNPs) were identified using the UnifiedGenotyper tool (version 2.8.1) and subsequently filtered using VariantRecalibrator (version 4.2.5.0), ApplyRecalibration, and VariantFiltration tools in the Genome Analysis Toolkit [27], implementing the following filter expression: QD < 2.0||MQ < 40.0||ReadPosRankSum < −8.0||FS > 60.0||HaplotypeScore > 13.0||MQRankSum < −12.5. For each gender group, SNP sites were examined: if an SNP site was predicted in an obese individual but not in the corresponding lean individual; the genotype of the lean individual at that site was set as homozygous, matching the reference genome; and the site was marked as a potential SNP between obese and lean broilers. If an SNP site appeared in both the obese and lean individuals, the genotype and nucleotide composition were confirmed based on GATK prediction results, with sites showing different nucleotides considered as SNPs between the two groups.

### 2.4. Statistical Analyses

All statistical analyses were performed using R software (version 4.0.2). Principal coordinates analysis was performed and visualized using two R packages (vegan and ggpubr), with the adonis *p*-value generated based on 999 permutations. The Kruskal-Wallis test, the Wilcoxon rank-sum test, and the *t*-test were employed to evaluate differences in various variables between groups. The resulting *p*-values were subjected to correction for multiple testing using the Benjamini-Hochberg procedure. The differential analysis and visualization of pathways between groups were conducted using the STAMP software (version 2.1.3) with default parameters. In contrast, the Kyoto Encyclopedia of Genes and Genomes (KEGG) and Gene Ontology (GO) enrichment analyses were performed for the chicken genome data using the OmicShare tool (https://www.omicshare.com/tools/, accessed on 1 October 2024). All graphical presentations were generated using R and the Adobe Illustrator environment.

## 3. Results

### 3.1. Breeding-Induced Alterations in Gut Microbiome

The average body weight of the lean-line broiler chickens was 2.14 ± 0.11 kg, comparable to that of the fat-line broiler chickens (2.02 ± 0.17 kg). However, the lean-line chickens demonstrated a significantly lower abdominal fat rate (1.16 ± 0.30%) compared to the fat-line (4.21 ± 0.35%; *p* < 0.001; Figure 1A). These results confirm that the selective breeding approach was effective in decreasing abdominal fat deposition without significantly impacting overall body weight, thereby enhancing broiler production efficiency.

To investigate the potential link between gut microbiota and the observed differences in fat deposition, we performed an excreta metagenome analysis on samples from the two broiler chicken lines using MetaPhlAn2 and HUMAnN2. Interestingly, despite the marked phenotypic differences, we did not observe significant differences in Shannon diversity or distinct clustering patterns in the principal coordinates analysis, suggesting minimal variation in microbiota diversity and overall community structure between the fat-line and lean-line chickens (Figure 1B,C). However, a closer examination revealed significant differences in the excreta microbial composition between these two broiler lines. *Bacteroides* (*B*.) and *Lactobacillus* (*L*.) were the predominant bacteria present in all chickens, accounting for 24.50% and 20.68% of the total microbiome, respectively (Figure 1D, Appendix A). Notably, the gut microbiome of lean-line chickens showed a higher relative abundance of *Lactobacillus* but a lower proportion of *Campylobacter*, *Gallibacterium*, and *Retroviridae* compared to the fat-line animals (Figure 1D). The major *Bacteroides* species identified included *B. barnesiae* (15.46%), *B. salanitronis* (3.12%), *B. coprophilus* (2.81%), *B. plebeius* (1.61%), and *B. coprocola* (1.50%), while the major *Lactobacillus* species were *L. salivarius* (6.67%), *L. helveticus* (5.04%), *L. crispatus* (4.81%), *L. johnsonii* (1.48%), and *L. vaginalis* (1.15%; Figure 1E, Appendix A).

These findings suggest that the selective breeding-induced alterations in abdominal fat deposition in broiler chickens were accompanied by specific changes in the composition of the gut microbiome, particularly in the relative abundances of key bacterial genera.

### 3.2. Abdominal Fat Deposition in Lean Chickens Was Linked to Altered Gut Microbiota Functions

To further investigate the potential association between the observed differences in gut microbiome composition and chicken abdominal fat deposition, we conducted a more detailed analysis of the differentially abundant taxa between the lean-line and fat-line chickens. We identified six bacterial species that were significantly less abundant in the lean-line compared to the fat-line chickens: *Aerococcus viridans*, *Avian endogenous retrovirus* EAV-HP, *Campylobacter coli*, *Corynebacterium casei*, *Enterococcus faecalis*, and *Lactobacillus gigeriorum* (*p* < 0.05). Additionally, the abundances of *Ruminococcaceae bacterium* D16 and *Turicibacter sanguinis* were found to be almost significantly decreased in the lean-line chickens compared to the fat-line (*p* = 0.051 and 0.052, respectively; Figure 2A). These differentially abundant taxa may represent key microbial signatures associated with the variations in abdominal fat deposition between the two broiler lines.

Given the observed breeding-induced alterations in gut microbiome structure and composition, we further explored the potential changes in the functional capacity of the excreta metagenome. Using the HUMAnN2 pipeline, we identified a total of 564 metabolic pathways represented in the chicken gut microbiomes (Appendix A). Interestingly, when comparing the lean-line and fat-line chickens, we found 18 pathways that were significantly enriched in the lean-line group. The enriched pathways in the lean-line chickens included those involved in the following: dTDP-3-acetamido-3,6-dideoxy-&alpha;-D-galactose biosynthesis, superpathway of geranylgeranyl diphosphate biosynthesis II, ubiquinol-6 biosynthesis from 4-hydroxybenzoate, superpathway of dTDP-glucose-derived O-antigen building blocks biosynthesis, taxadiene biosynthesis, adenosine nucleotides degradation II, stachyose degradation, isoprene biosynthesis I, purine nucleotides degradation II, superpathway of purine deoxyribonucleosides degradation, galactose degradation I, D-galactose degradation V, pyruvate fermentation to acetate and lactate II, superpathway of L-aspartate and L-asparagine biosynthesis and seleno-amino acid biosynthesis (*p* < 0.05; Figure 2B, Appendix A). In contrast, pathways related to mono-trans, poly-cis decaprenyl phosphate biosynthesis, L-isoleucine degradation I and 2-methylbutanoate biosynthesis showed the opposite trend, with higher abundances in the fat-line chickens (*p* < 0.05; Figure 2B, Appendix A). These findings suggest that the selective breeding-induced alterations in abdominal fat deposition in broiler chickens were accompanied by specific changes in the functional capacity of the gut microbiome. The differential enrichment of these microbial metabolic pathways may contribute to or reflect the divergent fat metabolism and deposition patterns observed between the lean-line and fat-line chickens.

### 3.3. Genomic Variations in Inbred Chickens Contributed to Alterations in Lipid Transport and Metabolism

To further elucidate the genomic basis underlying the observed differences in abdominal fat deposition between the lean-line and fat-line chickens, we conducted whole-genome resequencing of individuals from both populations. The sequenced data had an average size of 14.33 ± 1.01 Gb and a GC content of 42.18 ± 0.24%. We mapped these data to the *Gallus gallus* reference genome to identify SNPs in the lean-line and fat-line chickens. The identified SNPs were distributed across the chicken genome, with the exception of chromosomes 29, 34, 35, 36, 37, 38, and the sex chromosome W, which did not harbor any of the detected variants. The distribution of SNPs was uneven, with chromosomes 1 and 16 possessing the higher and lowest number of SNPs, respectively. Despite this sporadic distribution pattern, the overall SNP density calculations revealed no significant preference for specific chromosomes. In total, we identified 4241 SNPs located within the functional regions of the chicken genome, with the majority (4187 sites) situated within predicted protein-coding sequence. The remaining 54 SNPs were found in various non-coding RNA molecules, including small nucleolar RNAs, small nuclear RNAs, ribosomal RNAs, transfer RNAs, and other miscellaneous RNA species (Figure 3A; Appendix A). The presence of these SNPs within both coding and non-coding genomic regions suggests that the observed genetic variations may have the potential to impact a diverse range of cellular processes and regulatory mechanisms.

To understand the potential functional implications of these genetic variations, we performed pathway enrichment analysis on the genes harboring the identified SNPs. Our results revealed that these genes tend to cluster within six KEGG categories, predominantly involved in various metabolic functions, including lipid metabolism, amino acid metabolism, and energy metabolism (Figure 3B). This suggests that the observed genetic variations could have significant impacts on key metabolic processes underlying the distinct abdominal fat deposition phenotypes. Further enrichment analysis highlighted specific pathways related to the metabolism of vitamins, lipids, and glycans as being significantly enriched among the SNP-harboring genes (Figure 3C, Appendix A). Moreover, the GO-based functional enrichment analysis revealed that these genes were primarily associated with biological processes involved in lipid transportation and metabolism, as well as the regulation of brown adipocyte differentiation (Figure 3D).

Collectively, these findings provide valuable insights into the genetic mechanisms underlying the divergent abdominal fat deposition phenotypes observed between the lean-line and fat-line chickens. The identified genomic variations, particularly those associated with lipid-related metabolic pathways and processes, may play a crucial role in modulating the synthesis, transport, and overall metabolism of lipids, thereby contributing to the selection and development of the divergent adiposity phenotypes in these chicken populations.

## 4. Discussion

The deposition and accumulation of body fat in humans and animals is a complex, multifactorial process, influenced by a diverse array of genetic, dietary, environmental, lifestyle, and metabolic factors [28]. In the present study, we sought to explore the intricate connections between the gut microbiota and genomic variations in broiler chickens selectively bred for contrasting abdominal fat deposition phenotypes. By investigating these associations, we aimed to gain deeper insights into the underlying mechanisms that contribute to the divergence in fat deposition observed between the lean-line and fat-line broiler populations.

The intricate relationship between gut microbiota and body fat deposition has been reported previously, revealing that the composition and diversity of the gut microbial community can substantially impact fat deposition and distribution within the host organism [29]. Our results identified notable disparities in the composition and functionality of gut microbiota between the two distinct chicken lines, which had been selectively bred for either high or low abdominal fat deposition. A key finding was the marked increase in the abundance of *Lactobacillus* in the gut of chickens with reduced abdominal fat levels, suggesting a potential correlation between the intestinal *Lactobacillus* population and the host’s fat deposition processes [30]. While our understanding of this area continues to evolve, there is growing evidence to indicate that intestinal *Lactobacillus* may regulate fat metabolism and deposition through multiple pathways. Firstly, *Lactobacillus* may influence fat deposition by altering the host’s energy metabolism, which can in turn impact the intake and utilization of energy [19]. Secondly, the intestinal *Lactobacillus* population may affect fat deposition by modulating the colonic environment, thereby regulating the absorption and utilization of nutrients by the host [31]. Finally, some *Lactobacillus* strains may directly or indirectly produce bioactive metabolites, such as short-chain fatty acids, which can serve as energy sources for enterocytes and have the potential to influence the host’s energy metabolism and fat deposition [32]. In practical terms, there have been instances where broiler chicken productivity has been enhanced by regulating lipid metabolism through dietary supplementation with *Lactobacillus* [33,34]. Thus, our findings add to a growing body of evidence indicating the significance of specific gut microbiota, such as *Lactobacillus*, in regulating fat deposition and metabolic processes. Furthermore, we discerned differences in the functional gut metagenome between the fat-line and lean-line chickens. Notably, the gut microbiota of lean chickens exhibited significant enrichment in pathways related to biosynthesis, degradation, and metabolism, which is likely part of the mechanism for reducing fat deposition in this broiler line.

Both the existing evidence from the literature and the findings of this study support the close association between gut microbiota and fat deposition. This suggests that proactively manipulating the gut microbiota, especially during the breeding process, could potentially impact fat accumulation in broiler chickens. However, in this study, the gut microbiota of the two chicken lines were not proactively manipulated through probiotic supplementation or other dietary intervention strategies, and they were maintained under the same environmental conditions. Therefore, another likely reason for the divergent abdominal fat deposition phenotypes of the two lines is the specific genomic variations driven by the successive selective inbreeding process. Our research findings substantiate this notion, as we observed significant genomic differences between the fat-line and lean-line broilers, with a biased distribution of SNPs in genes associated with lipid synthesis and metabolism. For instance, SNPs were detected in the *APOB* gene (ENSGALG00000016491) that encodes apolipoprotein B, which is responsible for facilitating fat deposition by transporting excessive very low-density lipoprotein and low-density lipoprotein particles that deposit in tissues, leading to fat accumulation [35]. Moreover, the *PPARG* gene (ENSGALG00000004974) encodes the peroxisome proliferator-activated receptor-gamma, a well-recognized primary regulatory factor in adipocyte formation that promotes stem cell differentiation into adipocytes and regulates the synthesis and deposition of fat in mature adipocytes [36]. Additionally, *PLIN3* (ENSGALG00000002083) and *PLIN4* (ENSGALG00000021972) encode perilipin 3 and 4, which play a pivotal role in regulating the formation and metabolism of lipid droplets. They are responsible for shielding lipids within the lipid droplets from excessive metabolism or release during the fat deposition process [37]. However, this function can be dual, serving both protective and potentially promoting roles in fat deposition under different circumstances. The variation in SNP patterns between the fat-line and lean-line chickens may also translate to physiological changes within the host gastrointestinal environment, such as alterations in host secretions (e.g., gastric acid, bile, mucus), physical conditions (e.g., pH, oxygen levels), host immunity system, and efficiency in nutrient absorption and metabolism. These physiological changes may together contribute to remodeling the gut microbiome, feeding behavior, or other metabolic activities, particularly during the breeding stage of broiler chickens.

Drawing from our findings, we noted functional shifts in both the gut microbiota and the broiler chicken genome, primarily geared towards lipid and energy metabolism. Given our research context, it is reasonable to infer that the changes in the gut microbiota occur in tandem with host genomic variations. A close interaction exists between the gut microbiota and the host, where the intestinal microbiota regulates host health by assisting in digestion, regulating the immune system, and producing essential and bioactive metabolites through feedback mechanisms, while the host’s genetic factors and physiological status play a role in shaping its composition and function. This mutual interaction is integral to maintaining gut health and overall physiological balance. A previous study found that the gut microbiota composition and function are influenced by the host genome, potentially leading to its adaptive evolution to meet the host’s survival needs under environmental pressures [38].

Our findings lend support to this hypothesis of host-microbiome co-evolution, where shifts in certain microbial populations within the gut microbial community could be an adaptive strategy for more efficient regulation of energy metabolism and fat deposition to diverse host environments. For instance, our study observed an increased intestinal abundance of *Lactobacillus*; a decreased abundance of *Campylobacter coli*, *Corynebacterium casei*, and *Enterococcus faecalis*; and a differential abundance in multiple lipid metabolism-related pathways in the lean-line chickens. These microbial community changes may have ultimately contributed to the lower abdominal fat levels observed in these chickens. Similarly, the preferential distribution of SNPs in metabolism-related genes among the lean-line broilers could serve a similar adaptive purpose, potentially enhancing the host’s ability to regulate energy metabolism and fat deposition.

Such bidirectional adaptations between the host genome and gut microbiota not only exemplify host-microbiome co-evolution but also offer new insights into enhancing host health and optimizing poultry breeding and production efficiency. These complex interactions present opportunities for developing targeted strategies, such as selective breeding or microbiome manipulation, to modulate the gut-genome axis and improve fat deposition and other desirable traits such as meat production and quality in broiler chickens.

## 5. Conclusions

In summary, our findings confirmed that selective breeding can effectively reduce abdominal fat deposition in chickens. Notably, the lean-line chickens exhibited a markedly different intestinal microbial composition compared to the fat-line chickens. These compositional shifts in the gut microbiome were accompanied by alterations in microbial functional pathways, particularly those involved in energy metabolism and nutrient utilization. Furthermore, the selective breeding process led to specific SNPs in the lean broiler line, primarily in genes associated with energy and lipid metabolism. These genetic changes appear to have interacted synergistically with the observed alterations in gut microbiome composition, suggesting that both host genetic and microbial factors played key roles in regulating abdominal fat deposition.

Future research should focus on further elucidating the complex interactions between host genetics and the gut microbiome in modulating fat metabolism and deposition. Investigating the potential for probiotic, prebiotic, and other microbiome-targeted interventions, in combination with selective breeding strategies, may unveil new opportunities to optimize poultry growth, meat quality, and overall production efficiency.

## Figures and Tables

**Figure 1 animals-14-03174-f001:**
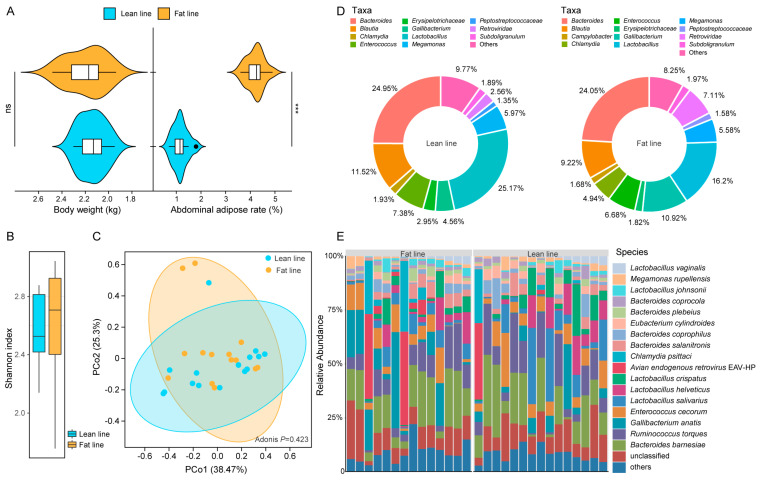
Comparative analysis of gut microbiome in two broiler chicken lines. (**A**) Comparison of body weight and abdominal adipose rate between chickens from fat and lean lines. Sample size: fat-line (n = 14); lean-line (n = 15). Statistical significance is indicated by *** *p* < 0.001; ns means non-significant. (**B**,**C**) Shannon diversity index and principal coordinates analysis (PCoA; Bray-Curtis distance) of gut microbiota of the two broiler lines. Significant differences between the two broiler lines were assessed by the Wilcoxon rank-sum test and the Adonis test. (**D**,**E**) Composition of excreta microbiota at different taxonomic levels. Taxa with relative abundances below 1% at the specific taxonomic level are grouped as “others”.

**Figure 2 animals-14-03174-f002:**
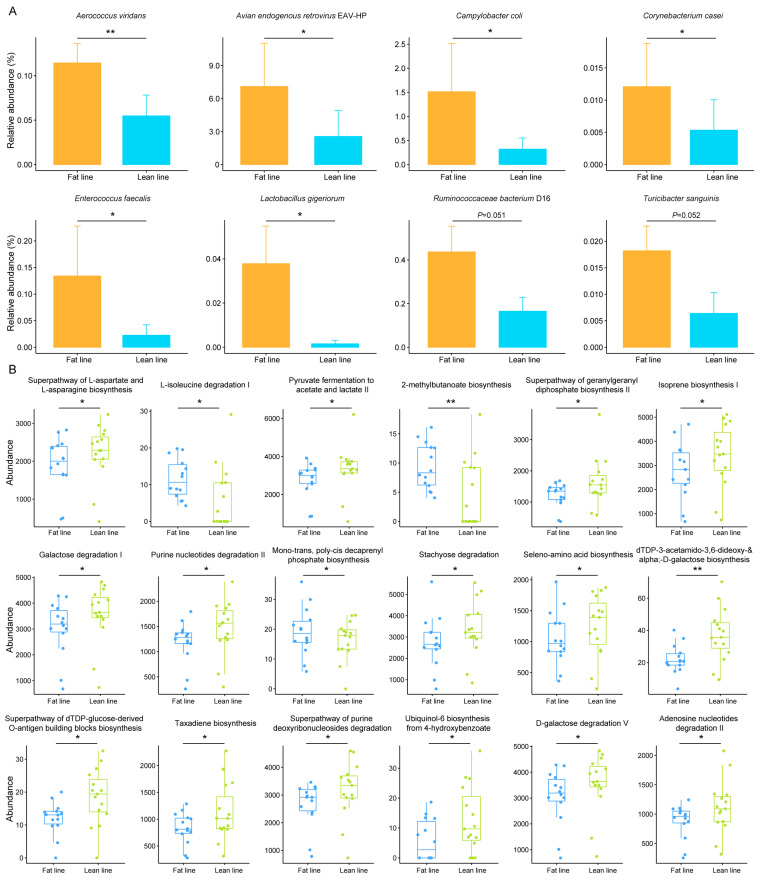
Differential taxonomic and functional metagenomes between fat-line and lean-line chickens. (**A**) Key differential microbial species. Error bars represent SEM. (**B**) Differential metabolic pathways. * *p* < 0.05; ** *p* < 0.01.

**Figure 3 animals-14-03174-f003:**
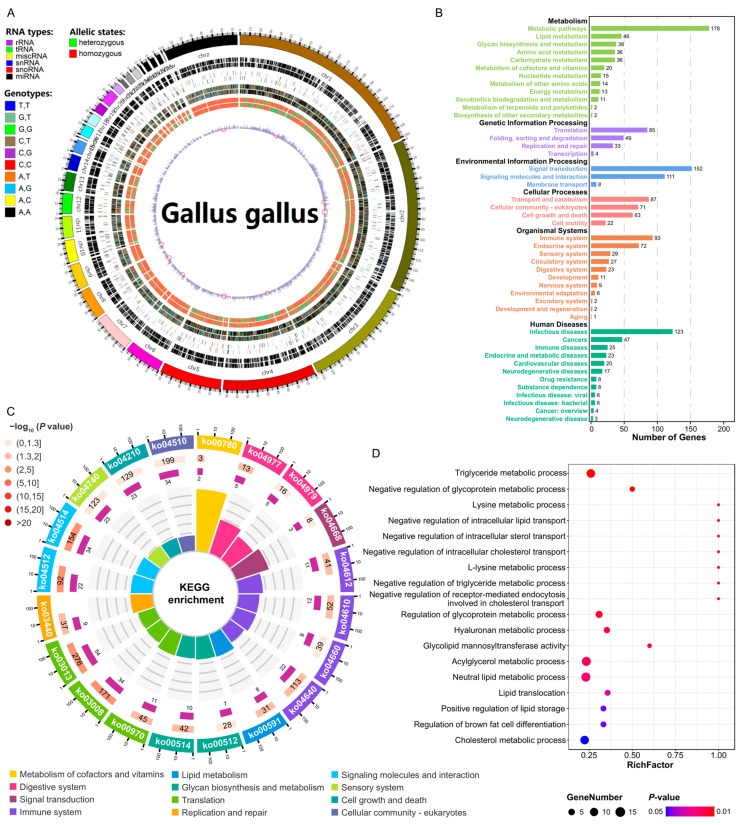
Whole-genome sequencing of fat-line and lean-line chickens. (**A**) Overview of chicken genome assembly and annotation. The outermost circle represents chromosomes with single nucleotide polymorphisms (SNPs). The innermost circle shows SNP density (number of SNPs per 1000 bases), with red regions indicating low SNP density. The middle circles (from outer to inner) represent annotations for coding sequences (CDS), RNA, and SNPs, respectively. (**B**) Kyoto Encyclopedia of Genes and Genomes (KEGG) enrichment pathways for genes carrying SNPs. Horizontal bars represent the number of genes enriched in each pathway. (**C**) Top significantly enriched KEGG pathways for genes carrying SNPs. The outermost circle specifies the enriched KEGG pathways and the scale for gene count, with functional grouping by color. The first and second inner circles displays the total and enriched gene counts for each KEGG pathway, respectively, with color intensity of the bar indicating the −log10 *p*-value, where longer bars denote a higher gene count. The innermost circle illustrates the RichFactor value for each KEGG pathway, corresponding to the relative proportion of enriched genes within the specific pathway, with gray grid lines at 0.1 intervals. (**D**) Top significantly enriched Gene Ontology (GO) terms for genes carrying SNPs. The color and size of each dot represent the *p*-value and gene count of the respective GO terms, respectively.

**Table 1 animals-14-03174-t001:** The nutrient values of chicken feed.

Nutritive Value	
Crude protein (%)	14.2
Metabolic energy (ME) (kcal/kg)	2745
Digestible amino acids (%)	
Arginine	0.13
Isoleucine	0.41
Lysine	0.61
Methionine	0.57
Methionine & cystine	0.45
Threonine	0.27
Tryptophan	0.18
Valine	0.51

## Data Availability

Datasets generated in this study have been deposited in the GenBank database (accession number: SRP083441).

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
