# Peer review of "Gut Microbiome-Host Genetics Co-Evolution Shapes Adiposity by Modulating Energy and Lipid Metabolism in Selectively Bred Broiler Chickens"

_animals, 2024, doi:10.3390/ani14223174_

Round 1
Reviewer 1 Report
Comments and Suggestions for Authors
The manuscript by Gao and colleagues focuses on gut microbiota and host genetic role in adiposity. They choose two broilers line selected for low and high fat accumulation.
There are few issues that need attention:
1. Are these lines commercial? What are the nutrient requirements for them? If they are not commercial, why NRC recommendation was used? Also line 114 indicate use NRC guidelines while line 129 indicates that Arbor Acre guidelines was used. Which one have you followed? Are these chicks Arbor Acres?
2. Please provide analyzed values for nutrient in Table 1 as well.
3. The use of feces is questionable. Why not ileum content or cecal content. Feces are not representative of cecal content unless you confirmed presence of cecal droppings in your samples. Additionally, chicken have excreta not feces.
4. Lines 178-180, STAMP only allows for visualization of results. What software was used to get the data for STAMP?
5. Some of the figures are too small and the text is not readable.
6. Sequencing raw data should be deposited in public databases.
Author Response
Comments from reviewer 1:
The manuscript by Gao and colleagues focuses on gut microbiota and host genetic role in adiposity. They choose two broilers line selected for low and high fat accumulation.
There are few issues that need attention:
Comments 1: Are these lines commercial? What are the nutrient requirements for them? If they are not commercial, why NRC recommendation was used? Also line 114 indicate use NRC guidelines while line 129 indicates that Arbor Acre guidelines was used. Which one have you followed? Are these chicks Arbor Acres?
Response 1: Thank you for the meticulous review, your professional insights have greatly improved the quality of our manuscript. The revised sections are marked in red in the revised manuscript. We apologize for the mistake of incorrectly applying the protocol of another parallel experiment in this manuscript. In fact, this study follows the Arbor Acres guidelines. The corresponding section in the revised manuscript has been changed to: “Throughout various life stages, the chickens were provided with different diets as per the Arbor Acres Plus parent stock breeder management guide and nutrition specifications (http://en.aviagen.com/). During the sampling period (ages ranging from 37 to 40 weeks), the chickens were subjected to feed restriction on a standard diet comprising 14.2% crude protein and 2745 kcal/kg of metabolic energy. Table 1 details the nutritive content of the feed and the digestible amino acid supplementation at the time of sampling. Through the above rigorous quality control measures, we ensured the consistency of feed components and quality, thereby effectively eliminating potential interference from feed on the results of intestinal microbiome diversity studies.” (revised manuscript Line 128-136)
Comments 2: Please provide analyzed values for nutrient in Table 1 as well.
Response 2: Thank you again for your thorough review and kind reminder. We have used the nutrient values as the revised Table 1.
Table 1. The nutrient values of chicken feed.
|
Nutritive value |
|
|
Crude protein (%) |
14.2 |
|
Metabolic energy (ME) (kcal/kg) |
2745 |
|
Digestable amino acids (%) |
|
|
Arginine |
0.13 |
|
Isoleucine |
0.41 |
|
Lysine |
0.61 |
|
Methionine |
0.57 |
|
Methionine & cystine |
0.45 |
|
Threonine |
0.27 |
|
Tryptophan |
0.18 |
|
Valine |
0.51 |
Comments 3: The use of feces is questionable. Why not ileum content or cecal content. Feces are not representative of cecal content unless you confirmed presence of cecal droppings in your samples. Additionally, chicken have excreta not feces.
Response 3: Thank you for your valuable comments. We appreciate your feedback, especially concerning the use of feces as our sample type. We would like to clarify our rationale and address your concerns. We chose “feces” as our sample type because it can represent the entire gut microbiome, not just specific segments. Previous research has shown that the microbial community in feces reflects the overall gut ecosystem, which is particularly relevant in poultry studies (such as PMID: 39150792, PMID: 37344888, PMID: 30728470, etc). There is indeed an increasing focus on the spatial variation in the composition in different sections of the chicken or other animals’ intestine. Thus, we acknowledge the importance of including cecal content for a comprehensive analysis. In our future studies, we plan to optimize our sample collection methods to ensure the inclusion of cecal contents, thereby enhancing the representativeness and depth of our findings. Finally, we have noted the distinction between excreta and feces. In the final version of our manuscript, we will use the more appropriate term “excreta” to accurately reflect the nature of chicken waste.
Comments 4: Lines 178-180, STAMP only allows for visualization of results. What software was used to get the data for STAMP?
Response 4: Thank you for your careful reminder. Line 151-153 of the manuscript mentions, "Functional annotation was achieved using the HUMAnN2 pipeline..." Our previous statement in lines 178-180 was problematic. It has now been revised as " The differential analysis and visualization of pathways between groups were conducted using the STAMP software (version 2.1.3) with default parameters." (revised manuscript Line 199-201)
Comments 5: Some of the figures are too small and the text is not readable.
Response 5: Thank you for your reminder. We initially submitted the manuscript with figures inserted into a Word document using the journal's template in the submission system. For this revised version, we will submit vector images.
Comments 6: Sequencing raw data should be deposited in public databases.
Response 6: Of course, our data has been uploaded to the public database. It has been appropriately referenced in the corresponding locations within the original text.
Data Availability Statement: Datasets generated in this study have been deposited in the GenBank database (accession number: SRP083441). (revised manuscript Line 475-476)

Reviewer 2 Report
Comments and Suggestions for Authors
This study is designed to investigate the effects of gut microbiome-host genetics co-evolution on adiposity and the related mechanism using a unique divergent lines of broiler chickens selected for abdominal fat weight. As this study tried to address an issue related to abdominal fat deposition in broiler chickens, it may have an important application in broiler industry. This study is novel, the method of analysis is creative, the results are interesting, and the data provide a substantial evidence for the conclusions. I recommend acceptance of this manuscript for publication after the following concerns are addressed:
1. There is still room to improve certain aspects of this article. For instance, the level of detail in experimental design and data analysis could be further enhanced to ensure the reproducibility and generalizability of the results. Additionally, the discussion on future research directions is somewhat brief and could benefit from a more in-depth consideration of how these findings can be practically applied in poultry production practice. Overall, this is an article of significant practical and theoretical value, though there is room for improvement in the elaboration of details and practical applicability.
2. Introduction: the authors may add more background and significance of selective breeding applications in agriculture.
3. Materials and methods: please describe the experimental samples in detail, including the breed, age, gender of the chickens, and other relevant information. Explain the specific methods of SNP analysis, including data sources, analytical tools, and selection criteria. Describe the composition of feed and quality control methods to exclude the influence of feed on the experimental results. Briefly explain the ethical considerations and handling methods for animals during the experimental process.
4. Discussion: I suggest that the authors add some potential research directions, for instance, whether gut microbiota affects the productivity and meat quality of broilers.
5. There are also some spelling or formatting issues. For example, the genera and phyla of bacteria should be italicized.
Author Response
Comments from reviewer 2:
This study is designed to investigate the effects of gut microbiome-host genetics co-evolution on adiposity and the related mechanism using a unique divergent lines of broiler chickens selected for abdominal fat weight. As this study tried to address an issue related to abdominal fat deposition in broiler chickens, it may have an important application in broiler industry. This study is novel, the method of analysis is creative, the results are interesting, and the data provide a substantial evidence for the conclusions. I recommend acceptance of this manuscript for publication after the following concerns are addressed:
Comments 1: There is still room to improve certain aspects of this article. For instance, the level of detail in experimental design and data analysis could be further enhanced to ensure the reproducibility and generalizability of the results. Additionally, the discussion on future research directions is somewhat brief and could benefit from a more in-depth consideration of how these findings can be practically applied in poultry production practice. Overall, this is an article of significant practical and theoretical value, though there is room for improvement in the elaboration of details and practical applicability.
Response 1: Thank you for your recognition and valuable suggestions. We will diligently revise this manuscript in accordance with your guidance to ensure its clarity, accuracy and overall quality. The revised sections are marked in red in the revised manuscript.
Comments 2: Introduction: the authors may add more background and significance of selective breeding applications in agriculture.
Response 2: Thank you for your suggestion. We have added the relevant literature to the Introduction section, and the reference sequence has been subsequently extended. The added sentences are as follows: “Selection breeding can not only directly improve fat deposition in broilers but also en-hance their overall health and growth efficiency through genetic improvement. For in-stance, Tan et al. comprehensively identifies genetic markers and regulatory mecha-nisms (such as the SOX6-MYH1s axis) involved in muscle development and yield in broiler chickens, offering new targets for selective breeding to enhance meat produc-tion efficiency and reduce myopathy [7]. Additionally, Shi et al. revealed the effects of selective breeding on skeletal muscle metabolism and meat quality through metabo-lomics technology, providing significant metabolic pathways and biomarkers for ex-ploring differences in meat quality between different broiler populations [8].” (revised manuscript Line 66-75).
Comments 3: Materials and methods: please describe the experimental samples in detail, including the breed, age, gender of the chickens, and other relevant information. Explain the specific methods of SNP analysis, including data sources, analytical tools, and selection criteria. Describe the composition of feed and quality control methods to exclude the influence of feed on the experimental results. Briefly explain the ethical considerations and handling methods for animals during the experimental process. The predicted SNPs were subjected to further bioinformatics analysis and validated with other experimental data.
Response 3: Thank you for your suggestions. We have made the modifications in the corresponding section.
- A total of 29 broilers were studied, comprising 14 from the fat-line (7 male and 7 female) and 15 from the lean-line (8 male and 7 female). (revised manuscript Line 125-126)
- Throughout various life stages, the chickens were provided with different diets as per the Arbor Acres Plus parent stock breeder management guide and nutrition specifications (http://en.aviagen.com/). During the sampling period (ages ranging from 37 to 40 weeks), the chickens were subjected to feed restriction on a standard diet comprising 14.2% crude protein and 2745 kcal/kg of metabolic energy. Table 1 details the nutritive content of the feed and the digestible amino acid supplementation at the time of sampling. Through the above rigorous quality control measures, we ensured the consistency of feed components and quality, thereby effectively eliminating potential interference from feed on the results of intestinal microbiome diversity studies. (revised manuscript Line 128-136)
Table 1. The nutrient values of chicken feed.
|
Nutritive value |
|
|
Crude protein (%) |
14.2 |
|
Metabolic energy (ME) (kcal/kg) |
2745 |
|
Digestable amino acids (%) |
|
|
Arginine |
0.13 |
|
Isoleucine |
0.41 |
|
Lysine |
0.61 |
|
Methionine |
0.57 |
|
Methionine & cystine |
0.45 |
|
Threonine |
0.27 |
|
Tryptophan |
0.18 |
|
Valine |
0.51 |
- This study was approved by the Special Committee on Scientific Research and Academic Ethics of Inner Mongolia Agricultural University (Approval No. NND2023107). (revised manuscript Line 117-119)
- For each gender group, SNP sites were examined: if an SNP site was predicted in an obese individual but not in the corresponding lean individual, the genotype of the lean individual at that site was set as homozygous, matching the reference genome, and the site was marked as a potential SNP between obese and lean broilers. If an SNP site appeared in both the obese and lean individuals, the genotype and nucleotide composition were confirmed based on GATK prediction results, with sites showing different nucleotides considered as SNPs between the two groups. (revised manuscript Line 186-192)
Comments 4: Discussion: I suggest that the authors add some potential research directions, for instance, whether gut microbiota affects the productivity and meat quality of broilers.
Response 4: Thank you for your suggestions. We appropriately mentioned strategies for changing fat deposition in broilers through regulating gut microbiota in the discussion and conclusion sections. Based on your recommendations, we also clarified some sentences. In fact, we have completed a study on improving digestion and absorption as well as enhancing immune regulation in broilers by feeding probiotics, which in turn increased meat production and quality. Related articles will be published.
Comments 5: There are also some spelling or formatting issues. For example, the genera and phyla of bacteria should be italicized.
Response 5: Thank you for your reminder, the formatting issues have been corrected accordingly.
